# Multiple-Rack Strategies Using Optimization of Location Assignment Based on MRCGA in Miniload Automated Storage and Retrieval System

**Miao He [1], Zailin Guan [1],\*, Chuangjian Wang [2] and Guoxiang Hou [3],\***

[1] State Key Laboratory of Digital Manufacturing Equipment and Technology, School of Mechanical Science and Engineering, Huazhong University of Science and Technology (HUST), Wuhan 430074, China
[2] Key Laboratory of Metallurgical Equipment and Control Technology, Wuhan University of Science and Technology, Wuhan 430081, China
[3] School of Naval Architecture and Ocean Engineering, Huazhong University of Science and Technology (HUST), Wuhan 430074, China
\* Correspondence: zlguan@hust.edu.cn (Z.G.); houguoxiang@163.com (G.H.)

**Abstract:** This paper aimed to introduce multiple-rack strategies in miniload automated storage and retrieval systems (AS/RSs), which included first fit (FF) and best fit (BF) assignment methods based on a matrix real-coded genetic algorithm (MRCGA) in the storage and retrieval process. We validated the probability occurrence of item sizes as a contributory factor in multiple-rack strategies, and compared their capacities, utilization of units and space by equal probabilities or the 80/20 law. According to the analytical methods, BF showed a reduction of more than 11.2% than FF on travel distance, and Type B-FF, Type B-BF and Type C-BF were better able to meet high-density requirements. These strategies provide diversified storage and retrieval solutions for the manufacturing and express delivery industry.

**Keywords:** multiple-rack strategies; probability occurrence of item sizes; first fit and best fit; matrix real coded genetic algorithm (MRCGA); miniload automated storage and retrieval system (AS/RSs)

## 1. Introduction

As the costs of manpower and land resources continue to rise [1], the traditional logistics warehousing technology gradually shows the disadvantages of low-space utilization and inconvenient operations. At the same time, under the influence of the acceleration of emergency support capacities of industries during the COVID-19 pandemic [2], higher requirements have been placed on automated warehouse technology [3]. Automated warehouse technology is the core of modern logistics technology, and its scope of application involves almost all industries [4].

Automated storage and retrieval warehouse systems (AS/RSs) satisfy high requirements for items promptly accessed and distributed, and a flexible rack strategy is an effective way to reduce costs and improve the space utilization of the warehouse. Generally, AS/RSs are composed of high-speed conveyors, dense racks, storage/retrieval (S/R) machines, input and output stations, and automatic control systems. As shown in Figure 1, a miniload AS/RSs take bins or cartons as the containers, realizing quick access to items of various sizes [5]. The stacker cranes fly back and forth on the lanes, loading and unloading the goods according to the control system's order lists. For example, stacker cranes move the cartons at the entrance (input station) of the lanes and store them into loaded units in racks. The racks are composed of metal brackets, partitions, and pallets, and they are widely used in intensive storage. Normally, a tunnel (two racks, and a stacker crane) can satisfy more than 100 items per hour for storage and retrieval operations [6].

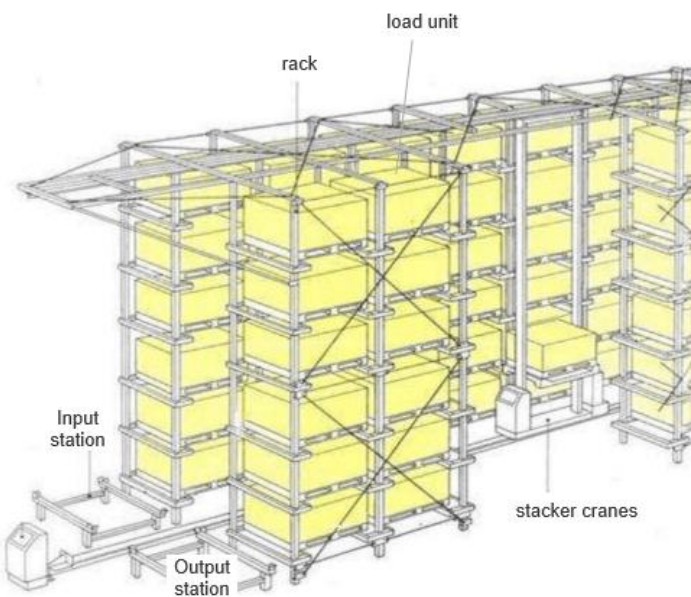

**Figure 1.** An example of Miniload AS/RSs.

The remainder of this paper is organized as follows. Section 2 provides a literature review, and the optimal solutions come from two main aspects, avoiding fragmentation and increasing rack capacity. In Section 3, three rack strategies are given. Then, their advantages and disadvantages using first fit (FF) and best fit (BF) methods based on MRCGA are expounded. In Section 4, their capacities and utilization are compared, considering the probability occurrence of item sizes by equal probabilities or the 80/20 law, and experimental results are discussed. Further more, we have explained the limitations of the study and the recommendation for future research.

## 2. Literature Review

AS/RSs have complex components and a large composition, and the modes of machine movement and transportation are diverse. Scholars at home and abroad have adopted many methods to improve their efficiency and practicability. Roodbergen et al. [7] provided an overview of the AS/RSs for the past 30 years, and a range of methods focusing on travel time estimation, storage assignment, and dwell-point location was explained. Jeroen P. van den Berg [8] considered the problem in selecting the dwell point position and machine idle time and presented analytic expressions for class-based and randomized policies. Hachemi et al. [6] solved the S/R assignment as a sequencing problem and used a step-by-step optimization method to gain the minimum double command (DC) time.

For improving system performance, Berglund et al. [9] developed an analytical solution procedure to minimize the expected path distance for the picker using a simplifying assumption. Manzini et al. [10] presented a new design and management approach by considering variable demand patterns. Banu et al. [11] developed an open queuing network-based software tool which estimated some important performance metrics in an SBS/RS system. Tony et al. [12] proposed a general mathematical method to minimize the waiting time by decomposing scheduled requests, such as location assignments and sequencing problems. Yener et al. [13] investigated the effectiveness of designing warehouses to reduce travel distance and order picking time. Chung et al. [14] presented a two-stage assignment (clustering and assignment) to minimize the picking delays from traffic congestion and travel time. Tone et al. [15] presented analytical travel time models to handle the calculation of expected cycle time in automated vehicles storage and retrieval system (AVS/RS) with a multiple-tier shuttle vehicle.

For low-carbon emissions, Ali et al. [16] presented a methodology model aiming to minimize the total cost of greenhouse gas (GHG) efficiency, and an ant colony optimization

(ACO) and genetic algorithm (GA) were developed to validate the obtained results. Liu et al. [17] proposed a robust facility method to measure system performance under disaster conditions, and the result was validated, at the same time, it was superior to traditional approaches. Yang et al. [18] presented a mathematical function model to analyze the picking strategy of delivery operation, and the overall operating efficiency was improved. Li [19] proposed a robotic mobile fulfillment system (RMFS) using three modules (task assignment, path planning and traffic control), which showed a higher warehouse space utilization in high-density storage warehouses, and saved approximately 10% storage spaces on average.

The explored research studies were concentrated on several themes, as shown in Table 1. We compared studies in system features, operating modes, and objectives. These objectives included time spent looking for items and items prior to sorting, and energy consumption. On the whole, practical application cases were mainly studied from the following aspects:

1.  First, reducing the travel or expected time of single single command (SC) or dual command (DC). Azzi et al. [20] suggested a new model to estimate the travel time and conducted a new Monte Carlo simulation. Huaining et al. [21] proposed an optimization model which was aiming to short the time of the retrieval and storage operation by combining free search (FS) and amendment circle algorithm. Ngoc et al. [22] proposed an efficient combination algorithm which reduced the travel distance in AS/RSs;

2.  Second, improving the compartment allocation strategies. Peng et al. [23] presented a variable neighborhood search (VNS) algorithm to solve the large-sized item operations under shared storage in multi-shuttle AS/RS. They used random or nearest storage strategies, classified, or shared according to the original data and resources, such as item numbers, material types, sizes, weights, etc.;

3.  Lastly, improving the performance of the AS/RSs platform, such as better planning of scheduling operations. Tostani et al. [24] proposed a novel bi-level and bi-objective model which could offer better planning of operations. Tian et al. [25] proposed two continuous travel time models, such as a dedicated lift per job type and rack, and two models were validated by simulation and showed accurate results.

**Table 1.** The studies of optimization in AS/RSs.

| Literature | System Type | System Features | Operating Modes | Objectives |
|---|---|---|---|---|
| [8] | AS/RS | position where the S/R machine resides | Undefined | minimizing the expected time to the first operation after an idle period |
| [7] | AS/RS | dynamic scheduling and design | SC/DC | improving system performance of large computation times and finite planning horizons |
| [4] | AS/RS | accelerating/deceleration of the S/R machine | SC/DC | reducing the expected travel time |
| [20] | AS/RS | dual-shuttle | SC/DC | SC and DC travel times |
| [6] | AS/RS | unit-load location rule | DC | minimizing DC travel times |
| [26] | Miniload AS/RS | identical shelves which handle different widths cartons | Undefined | storage space utilization |
| [25] | AS/RS | multi-shuttle | Undefined | operational efficiency |
| [12] | Miniload AS/RS | dual shuttle crane | DC | minimizing the prioritized waiting time |
| [27] | AS/RS | single crane scheduling | SC/DC | a novel classification scheme |
| [16] | AS/RS | unit-load multiple-rack | DC | minimizing the cost of GHG efficiency |
| [28] | flow-rack AS/RS | a multi-deep rack and two machines | SC/DC | an analytical model for the performance evaluation and design |
| [5] | Miniload AS/RS | shuttle vehicles-type | Undefined | enhancing the buffering function of flexible storage and sorting operations |
| [25] | Split-platform AS/RS | 2 flexible lifts/2 racks | DC | DC travel time |
| [15] | AVS/RS | multiple-tier shuttle vehicles | SC/DC | an analytical travel time models |
| [19] | RMFS | high-density storage warehouses with limited space or high rental costs | DC | saving labor costs and achieve higher picking efficiency |
| This paper | Miniload AS/RS | Multiple rack design strategies | DC | reducing the fragmentation and increasing rack capacity |

In manufacturing enterprises, the daily input and output (I/O) goods include all kinds of materials from whole products and components to tiny spares. The material characteristics include multiple types, varieties, sizes, and tens of thousands of shapes. Meanwhile, no one strategy could perfectly solve all storage and retrieval problems.

The materials are loaded in containers, but for different shapes and sizes, their quick and efficient S/R operations are hard to realize. The containers, storing materials such as spares, components, and whole products, are sophisticated (such as the containers used in the auto manufacturing enterprises). Daily orders are uncertain and urgent. In this paper, we assumed that order could be predicted within a variation range. In limited physical spaces, the solutions are derived from two aspects:

1. Avoiding fragmentation. Tokola and Niemi [26] proposed a minimizing fragmentation method in a horizontal direction, reducing the gaps between cartons due to several times for input/output operations. When the gaps were too narrow to load any cartons, they were wastes of space. The horizontal direction was fully discussed in their paper, but the vertical direction was dismissed. In this paper, we used space utilization as one of the optimal indexes, discussing the utilization of space in various rack strategies. In addition, we added vertical direction as an important model parameter, and a unity and equal-depth model to reduce the computation amounts;

2. Increasing rack capacity. In order to increase rack capacity, the quantity of loaded units should be as large as possible, and various sizes of containers or cartons, multiple types of rack strategies were proposed. Rao and Adil [29] presented a class-based method using a modified version (an ABC curve) on turnover density. Jason et al. [30] presented an effective heuristic algorithm to locate products for a pick-and-pass system. Sunil et al. [31] presented a decision model according to various parameters such as total delivery time, total investment on each item and total cost. Chen et al. [32] proposed a hierarchical two-stage-exchange method to minimize the total dispersion degree in large-scale transshipment (commercial cars). Derhami et al. [33] presented a block-stacking method by bay depth, cross-aisle types and the number of aisles and cross-aisles in the beverage industry, and the resulting layout reduced operation costs by 10%. Ghomri et al. [28] proposed an analytical method that took into account various items' physical parameters, such as length, width, and depth. Extensive research is based on same-size containers and the same racks, and each unit is only loaded one container. Obviously, there are horizontal and vertical wastes of space. In our paper, the above-mentioned rack strategies were considered as the Type A model, and we added other models to increase the rack designs for various storage requirements, aiming to maximum use of racks by measuring their capacities and utilization.

Gaku and Takakuwa [5] presented a method to demonstrate bottlenecks in different layouts and take operation priorities and allocation rules into consideration in mini-load AS/RSs. Ouhoud et al. [34] proposed a continuous model to study configurations and analyzed various discrete distributions in horizontal and vertical movements in multi-aisle AS/RS. Zhang et al. [35] presented a dynamic stocking decision and two heuristics for small lots and generated more efficient picklists. On the other hand, the quantities of materials are different and increase the storage difficulties. Boysen et al. [27] presented a novel classification scheme to precisely define variety scheduling problems in single storage/retrieval machines. Based on these studies, we used equal probabilities and the 80/20 law [36] to simulate input lists and storage/retrieval operations, and to validate rack strategies by a matrix real-coded genetic algorithm (MRCGA), and the whole verification methods to compare the advantages and disadvantages of models in an efficient way.

To meet the needs of storing complexity, high utilization, and efficiency, we proposed multiple rack strategies for storage and retrieval assignment policy in miniload AS/RSs.

In this paper, our discussions mainly focused on avoiding fragmentation and increasing rack capacity in limited space in miniload AS/RSs. In our methods, the limited space

for racks was designed as a three-dimensional model, and the mathematical model was established as follows.

1.  Single-deep rack and stack design;
2.  All kinds of cartons are suitable for storage, no matter the size;
3.  Racks have same outline, means that they have same length, height and depth. At the same time, cartons have same depth as racks. In this way the space utilization is based on length and height (horizontal and vertical);
4.  Every carton's length is equal to its height;
5.  One rack has several lines, and each line has several loaded-units, and each loaded unit's length is equal to its height. In this paper, a rack has 10 to 40 locations per line, and 4 to 12 locations per column;
6.  Time costs on preparing storage and retrieval are handled as constants in travel process and could not be count in when calculating the travel distance. In the process of storage/retrieval operations, there are short time periods for preparing storage and retrieval. For example, from stacker cranes grab the cartons to start moving, or from stacker cranes load the cartons to start moving. Compared to travel times, these time periods are short. In addition, every DC operation has the same short time periods so that we set them as constants, and when we compare the travel distances in three rack strategies, these time periods are subtracted. Thus, we arrived at the conclusion that these time periods could not be counted on.

Table 2 summarizes the definitions of parameters and variables used in this paper.

**Table 2.** Table of notation.

| |
| --- |
| $L_{rack}$ Length of rack, all racks have same $L_{rack}$ |
| $H_{rack}$ Height of rack, all racks have same $H_{rack}$ |
| $L_{unit}$ Length of loaded-unit |
| $H_{unit}$ Height of loaded-unit |
| $L_{carton}$ Length of carton |
| $H_{carton}$ Height of carton |
| $Q_{rack}$ the total of racks |
| $Q_{unit}$ the total of units |
| $Q_{carton}$ the total of cartons |
| $T$: the total of types of unit sizes |
| $i$: index of rack, $i \in [1, Q_{rack}]$ |
| $j$: index of units, $j \in [1, Q_{unit}]$ |
| $m$: index of carton, $m \in [1, Q_{carton}]$ |
| $T_G$: the required time for grabbing cartons and ready to move in storage operation, here it is defined as a constant. |
| $T_L$: the required time for grabbing cartons and ready to move in retrieval operation, here it is defined as a constant. |
| $Q_{in}$: the quantity of stored cartons in daily order |
| $e$: index of stored cartons, $e \in [1, Q_{in}]$ |
| $Q_{out}$: the quantity of retrieval cartons in daily order |
| $f$: index of retrieval cartons, $f \in [1, Q_{out}]$ |
| $GEN$: the quantity of initial population in MRCGA, containing all chromosomes. |
| $k$: index of chromosome, $k \in [1, GEN]$ |
| $ITER$: the total of iteration |
| $d$: the index of iteration, $d \in [1, ITER]$ |

### 2.1. Type A Rack Strategy

In Type A rack strategy, one unit permits only one carton loading. To satisfy the permission, all rack units are big enough to all cartons, no matter the sizes of cartons are. In these situations, it is clear that there are gaps in units when the cartons are smaller than rack units (gaps are represented by blue arrows in Figure 2).

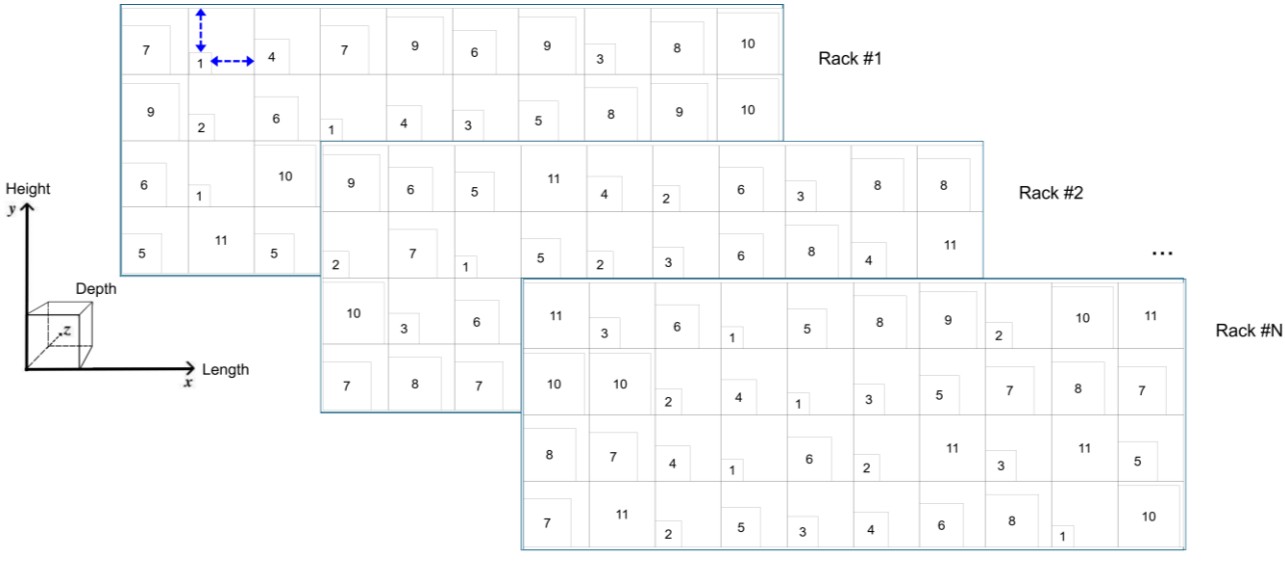

**Figure 2.** The side view of Type A racks.

The capacity of Type A is:

$$Q_{unit} = Q_{rack} \cdot \frac{L_{rack}}{L_{unit}} \cdot \frac{H_{rack}}{H_{unit}} \tag{1}$$

When rack units are loaded with cartons, we define the units utilization as $Utilization_{units}$, and space utilization in $x$-$y$ square (the square area is calculated as $L_{rack} \cdot H_{rack}$) as $Utilization_{xy}$.

$$Utilization_{\text{units}} = \frac{Q_{carton}}{Q_{unit}} \cdot 100\% \tag{2}$$

$$Utilization_{\text{xy}} = \frac{\sum\limits_{m=1}^{Q_{carton}} (L_{carton,m} \cdot H_{carton,m})}{Q_{unit} \cdot L_{unit} \cdot H_{unit}} \cdot 100\% \tag{3}$$

In Formula (2), $m$ is represented as the index of cartons, from 1 to $Q_{carton}$.

### 2.2. Type B Multiple Sizes Rack Strategy

The design of Type B aims to reduce the gaps in the Type A strategy. Considering that an appropriate carton is placed in an appropriately sized rack unit, ideally, the gap is totally avoided, as shown in Figure 3. There are 11 types of cartons, meanwhile, the sizes of the rack units have 11 types. It is obvious that the quantity of storage units is greater than that of Type A in the same limited space.

Setting the quantity of smallest unit racks is $Q_{rack,1}$, the second smallest is $Q_{rack,2}$, ... , and the biggest is $Q_{rack,T}$ ($T$ is the quantity of unit types, and $t$ is the index, from 1 to $T$), and $Q_{rack,1} + Q_{rack,2} + \ldots + Q_{rack,t} + \ldots + Q_{rack,T} = Q_{rack}$. Moreover, $L_{unit,t}$ and $H_{unit,t}$ is the unit's length and height of #$t$ size rack, and the capacity of Type B is:

$$Q_{unit} = Q_{rack,1} \cdot \frac{L_{rack}}{L_{unit,1}} \cdot \frac{H_{rack}}{H_{unit,1}} + Q_{rack,2} \cdot \frac{L_{rack}}{L_{unit,2}} \cdot \frac{H_{rack}}{H_{unit,2}} + \ldots + Q_{rack,T} \cdot \frac{L_{rack}}{L_{unit,T}} \cdot \frac{H_{rack}}{H_{unit,T}} \tag{4}$$

When the units are stored with cartons, we define $Utilization_{xy}$ as follow:

$$Utilization_{xy} = \frac{\sum\limits_{i=1}^{Q_{carton}} (L_{carton,i} \cdot H_{carton,i})}{Q_{rack} \cdot (L_{rack} \cdot H_{rack})} \cdot 100\% \tag{5}$$

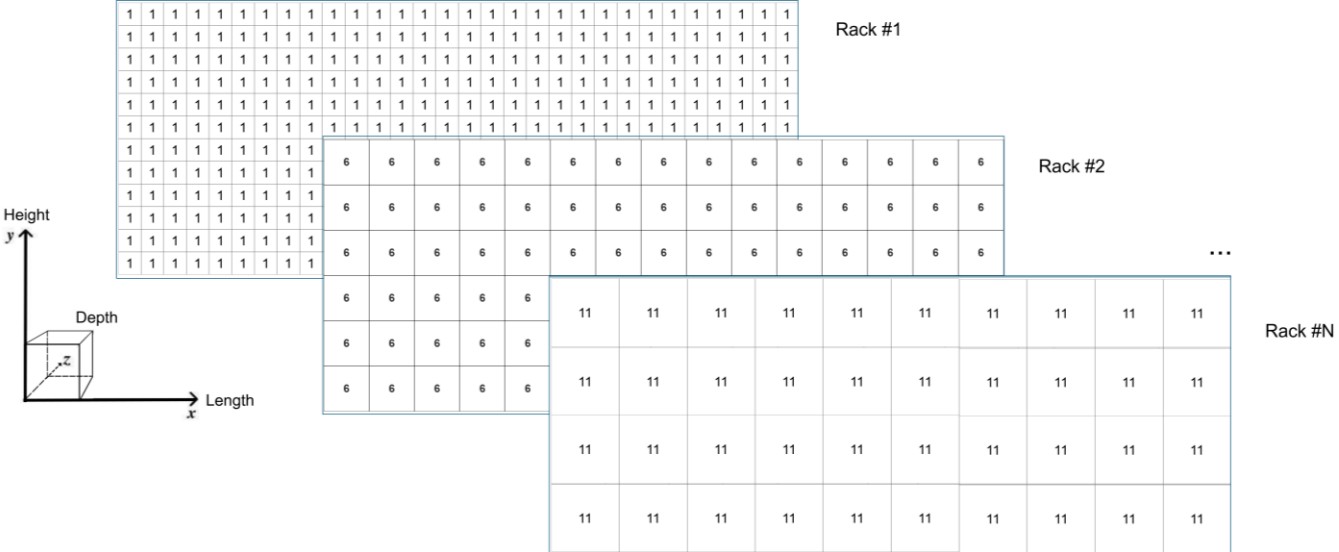

**Figure 3.** The side view of Type B racks.

### 2.3. Type C Multiple Cartons Rack Strategy

Nowadays, the racks are normally designed in same unit sizes, and it is simple for items' placement and transportation. In reality, as technology advances, containers (pallets or cartons) can share their units shown in Figure 4, and it is obvious that the containers or pallets are unified into one size.

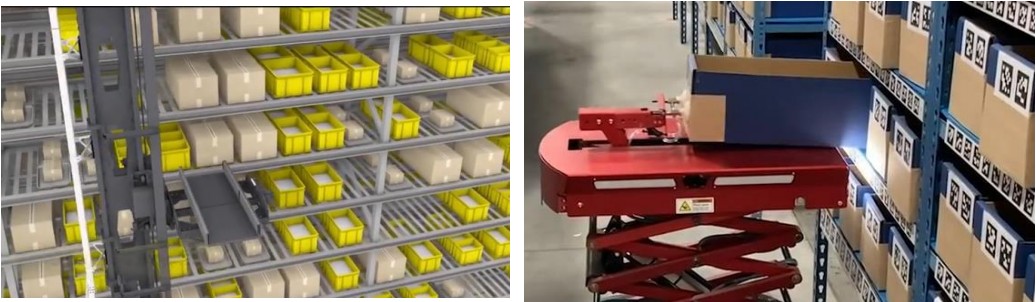

**Figure 4.** Examples of multiple items in one load unit.

In the Type C strategy, assuming that one unit could be stored several cartons, and space allows. As shown in Figure 5, one unit could load three size #1 cartons, or one size #1 and one size #6 carton.

The capacity of Type C is:

$$Q_{unit} = Q_{rack} \cdot \frac{L_{rack}}{L_{unit}} \cdot \frac{H_{rack}}{H_{unit}} \tag{6}$$

When the units are stored with cartons, we define $Utilization_{xy}$ as follow:

$$Utilization_{xy} = \frac{\sum_{i=1}^{Q_{carton}} (L_{carton,i} \cdot H_{carton,i})}{Q_{rack} \cdot (L_{rack} \cdot H_{rack})} \cdot 100\% \tag{7}$$

Using Formula (2), $Utilization_{units}$ could be more than 100% in Type C due to multiple cartons in one unit.

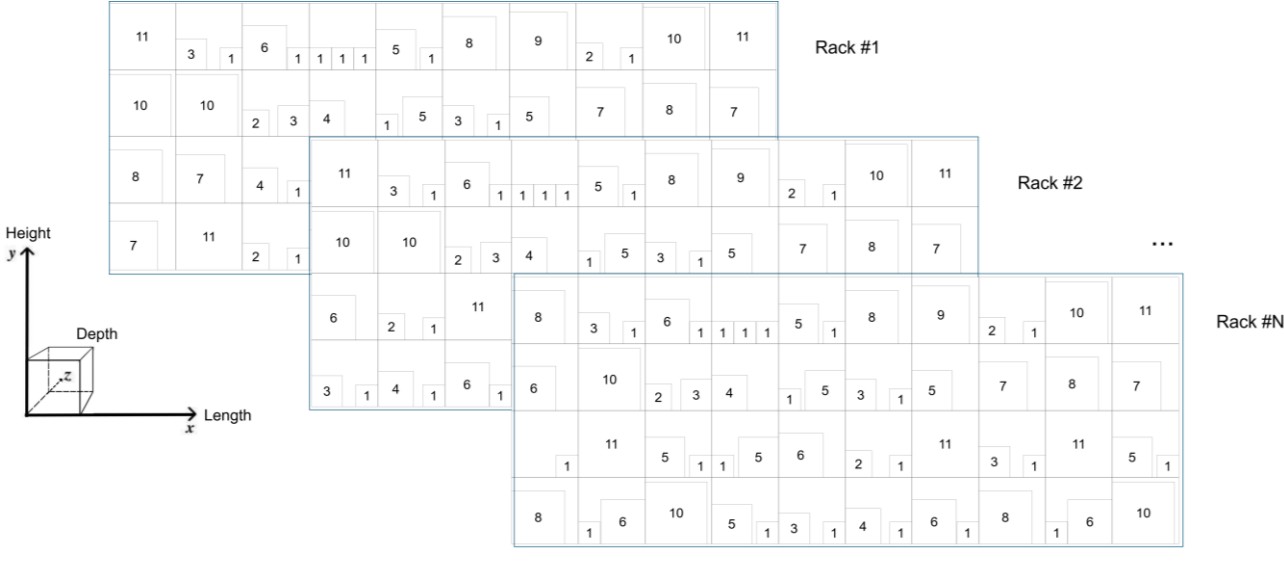

**Figure 5.** The side view of Type C racks.

### 2.4. FF and BF Location Assignment Methods

According to the locations of cartons, there are two methods usually used in the location assignments, First Fit (FF) and Best Fit (BF). As shown in Figure 6, every location is defined as their coordinates (ArrayIndex, LevelIndex). For example, the array 2, level 1 unit is (2,1), showed as arrow #1 and arrow #3, or (1,4) is showed as arrow #4 and arrow #2. Besides, the searching order is from left to right, then bottom to top. Using this searching rule, there are several locations are suitable for loading #6 carton, for example, (1,4), (2,1), (2,2), (2,3), (3,4), and (2,4) in Type C rack.

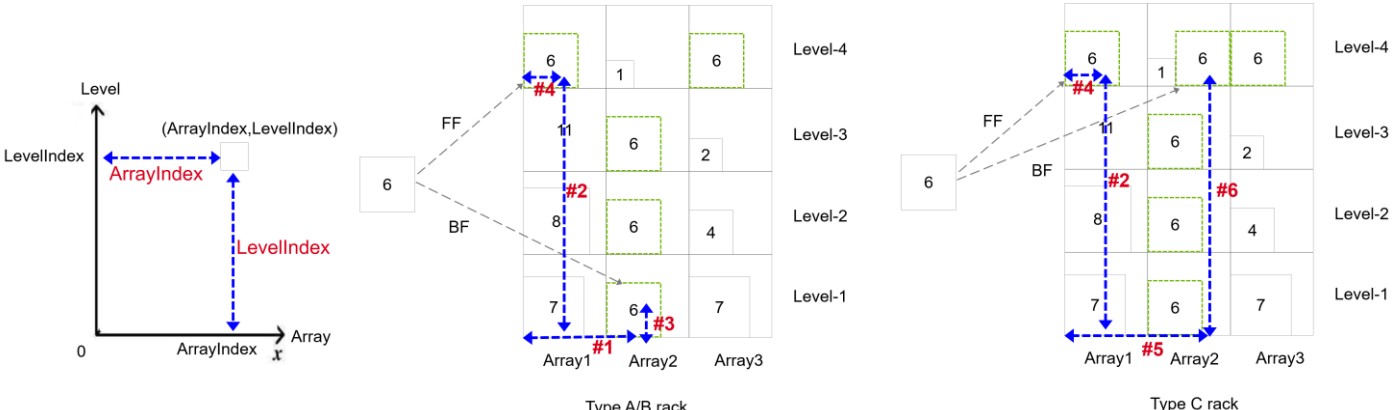

**Figure 6.** The BF and FF method in Type A/B/C models.

*FF* method chooses the first location as the load destination, thus the (1,4) is the final position coordinate, in array 1 and level 4. In Formula (8), the quantity of solutions is *K*, and *k* is their index.

$$FF = First\ (coordinate_1, \ldots, coordinate_k, \ldots, coordinate_K),\ k \in [1, K] \qquad (8)$$

*BF* method uses judgement function to choose the final solution, and it is given in Formulas (9) and (10). In Type A or Type B rack strategy, judgement function is represented by the least travel distance of all feasible solutions ($BF_{D,Dis\tan ce}$), for instance, we get (2,1) in Figure 6.

$$BF_{D,Dis\tan ce} = Min(\sqrt{ArrayIndex_j{}^2 + LevelIndex_j{}^2}),\ j \in [1, Q_{unit}] \tag{9}$$

In Type C rack strategy, judgement function has 1 or 2 steps.

Step 1, calculate the most suitable locations ($BF_{D,Gap}$, in Formula (10)). If $BF_{D,Gap}$ is no less than $L_{unit}$, meaning that there no shared unit for carton, then go to Step 2; or $BF_{D,Gap}$ gives a shared unit space for the loading carton, and function end.

Step 2, calculate $BF_{D,Dis\tan ce}$, using Formula (9).

For instance, we get shared unit (2,4), showed as arrow #5 and #6 in Figure 6, for #6 carton in Type C rack (in Figure 6).

$$BF_{D,Gap} = Min \sum_{j=1}^{Q_{unit}} (L_{unit,j} - L_{carton,j}),\ j \in [1, Q_{unit}] \tag{10}$$

In Formulas (9) and (10), $Q_{unit}$ is the quantity of units, and $j$ is the index of units.

### 2.5. Analysis of DC Using MRCGA

There are large amounts of storage and retrieval requests in daily production, and scholars develop some improved Genetic Algorithm (GA) algorithms for these S/R assignments in many research studies. GA has a wide range of applications and the characteristics of biological selection and heredity in nature [37]. In the process of iterations, it makes good use of its own crossover and mutation to find the optimal solution quickly, and has an excellent ability to search for solutions in the global range [8,38]. Li et al. [37] presented a greedy genetic algorithm for a new dynamic storage assignment problem. Hu et al. [39] proposed a yard sharing Non-Dominated Sorting Genetic Algorithm-II (NSGA-II) to ease container congestion in surplus storage space, and the analysis is conducted by weight coefficient and feasible distance. Peng et al. [38] presented a mixed integrated programming model and improve the standard NSGA-III by crane scheduling and relocating rate. In this paper, we propose a matrix real-coded genetic algorithm (MRCGA), which is based on matrix selection, crossover, and mutation operations for each generation of population individuals, and the process is shown in Figure 7. Each randomly generated matrix is regarded as a chromosome (feasible solution), and the initial population is a number of matrixes. This matrix coding method can not only reduce the computational workload, but also ensure the feasibility and legitimacy of each offspring in the process of crossover and mutation.

As shown in Figure 8, a DC operation is traced. First, stacker crane grabs the #1 carton from the input-port and transports it into the target location (using FF or BF method to get the target location), the location is showed as arrow #1 and #2. Then the machine moves to the #7 carton, the trajectory is showed as arrow #3, and grabs it and transports it to the output-port, the carton location is showed as arrow #4 and #5. In the whole process, $T_G$ and $T_L$ (defined in Table 2) is handled as constants and ignored in this paper, and the travel distance ($D_{travel}$) is the main factor in the R/S processes.

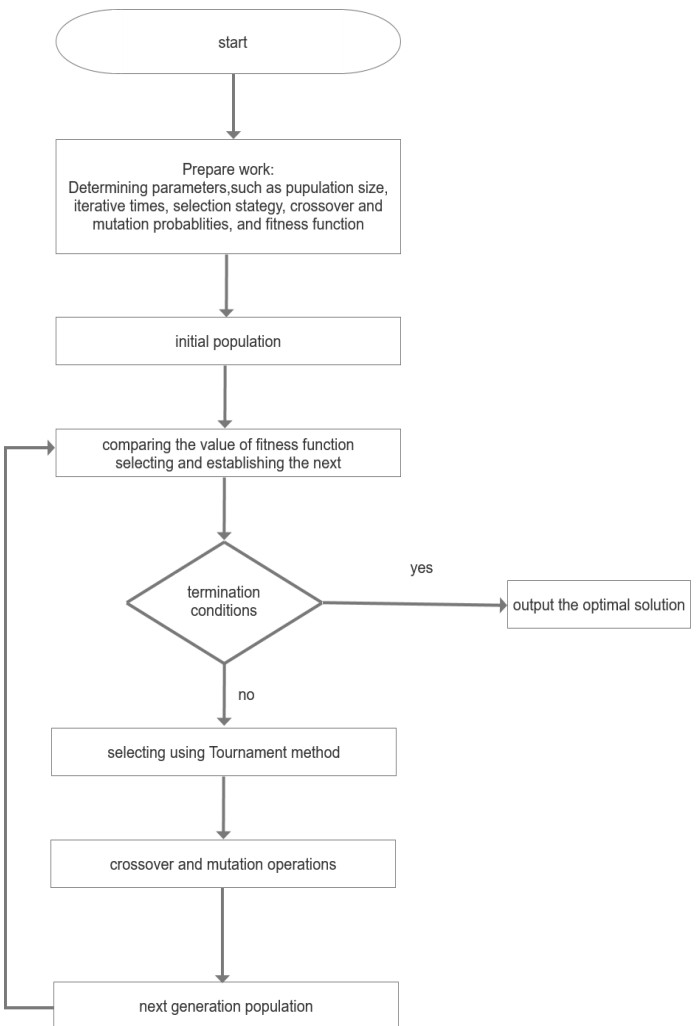

**Figure 7.** Flow chart of MRCGA algorithm.

In Formula (11), a double dual-command cycles contain a loading coordinate represented as $(ArrayIndex_e, LevelIndex_e)$, and a picking coordinate $(ArrayIndex_f, LevelIndex_f)$. The travel distance ($D_{travel}$) is showed in Formula (11).

$$D_{travel} = \sqrt{ArrayIndex_e^2 + LevelIndex_e^2} +$$
$$\sqrt{\left|ArrayIndex_f - ArrayIndex_e\right|^2 + \left|LevelIndex_f - LevelIndex_e\right|^2} \qquad (11)$$
$$+ \sqrt{ArrayIndex_f^2 + LevelIndex_f^2}$$

Normally, there is an amount of loading and picking requests in a period, and we proposed to optimize the combinations by MRCGA. Combining the loading and picking coordinates randomly, and single chromosome $Chr_k$ was generated as below. It is remarkable that the quantities of storage and retrieval could be unequal.

$$Chr_k = \begin{bmatrix} LoadingIndex_1 \ PickingIndex_1 \\ LoadingIndex_2 \ PickingIndex_2 \\ LoadingIndex_3 \\ \vdots \qquad\qquad \vdots \\ LoadingIndex_N PickingIndex_M \end{bmatrix} \qquad (12)$$

$D_{travel}$ is considered as the fitness function, and $D_{travel,chr_k}$ is the $D_{travel}$ of $Chr_k$. When $D_{travel,chr_k}$ is smaller, $Chr_k$ is more likely to be the optimized solution.

$$Fit_d = \min(D_{travel,chr_k}) \tag{13}$$

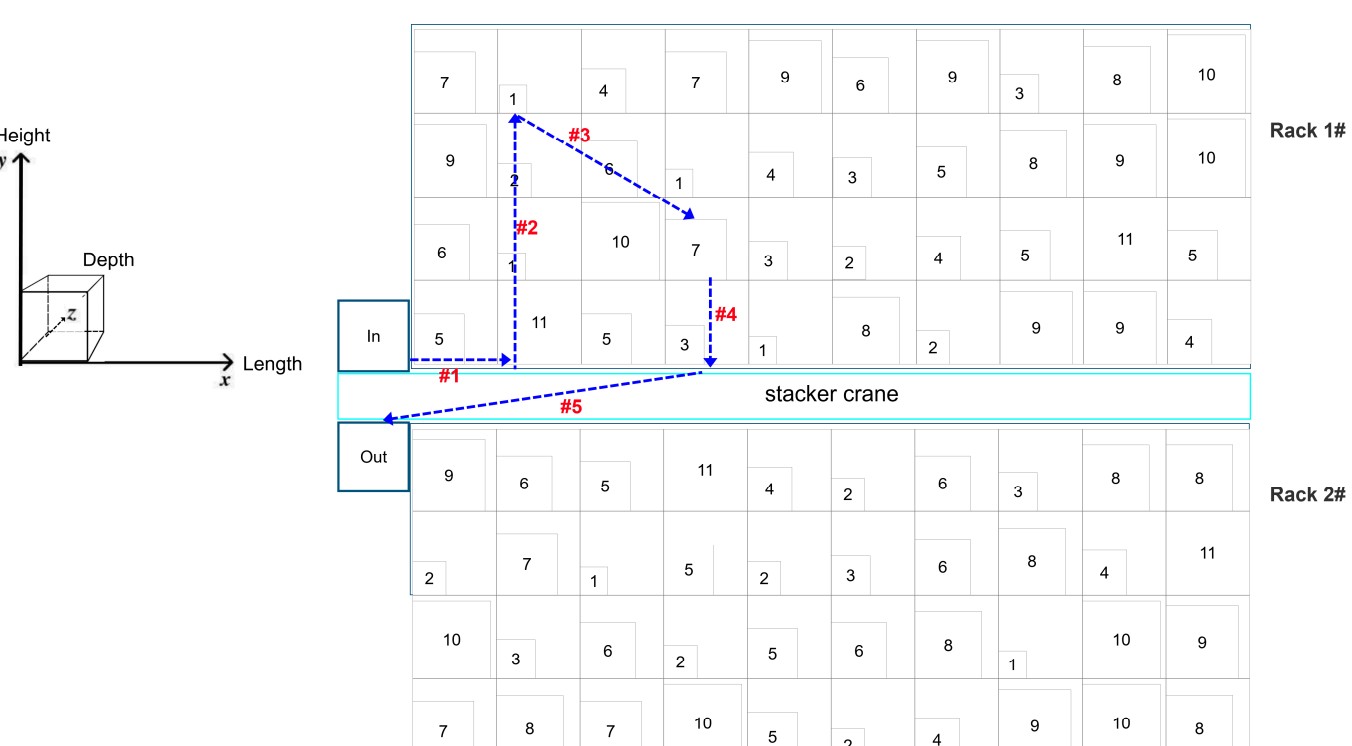

**Figure 8.** An example of DC in the Type A rack strategy.

In the selecting steps, we use tournament method to choose better parents chromosome by $Fit_d$, then the crossover and mutation probability ($cp$, $mp$) are setting as constants (such as $cp = 0.9$, $mp = 0.2$). When the number of iterations reaches the maximum setting ($ITER$, in Table 2), the optimal solution is output. The solution is the optimized location assignment for the S/R cartons.

## 3. Numerical Experiments

### 3.1. The Capacity and Utilization of Type A

In Type A, there is an advantage that it is simple to allocate the cartons when all the loaded-units are same sizes. Setting $Q_{rack} = 1$, $L_{rack} = 30$, $H_{rack} = 12$, $L_{unit} = 3$, $H_{unit} = 3$, and we get $Q_{unit} = 40$. Setting $Q_{in} = 40$, $Q_{out} = 0$, and 11 carton sizes (in Figure 9) are presented randomly.

As shown in Figure 9, 11 sizes are expressed as ($L_{carton}$, $H_{carton}$), such as #1 (1,1), #2 (1.2,1.2), #3 (1.4,1.4), . . . , #6 (2,2), . . . , #10 (2.8,2.8), #11 (3,3). When the probability occurrence of 11 sizes of cartons are nearly equal, more or less than 10%, and the rack is full of cartons, then we get that $Utilization_{units}$ is 100% and $Utilization_{xy}$ is 49.72% by Formulas (2) and (3). It suggests that even all units are loaded, there is nearly a half space could be improved, then we discuss the Type B and Type C strategies.

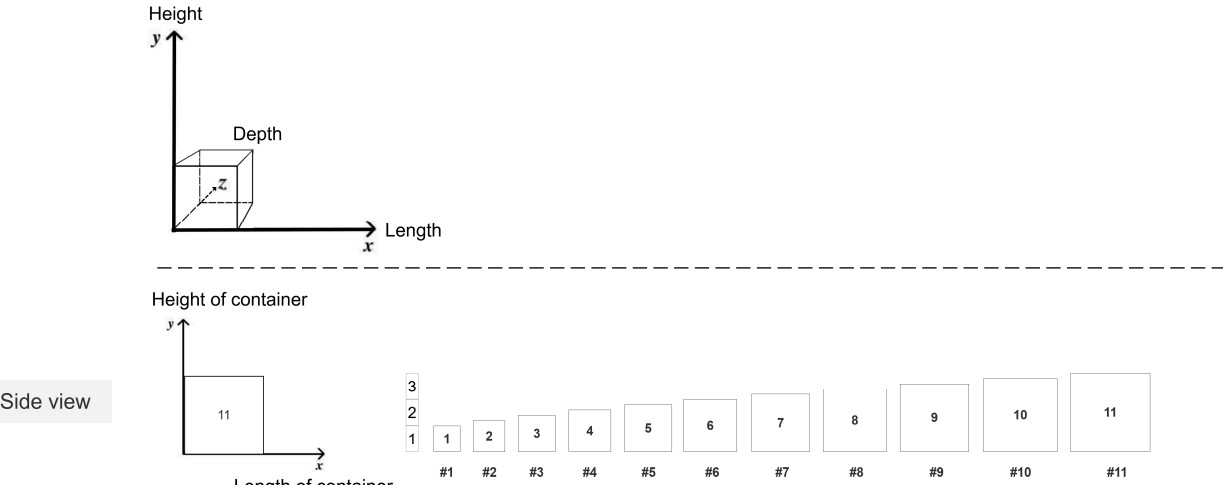

**Figure 9.** The 11 sizes of cartons.

### 3.2. The Capacity and Utilization of Type B

In Type B, we set $Q_{rack} = 11$, and the 11 rack-unit sizes are matched with the 11 carton sizes in Figure 9. The capacity is $Q_{unit} = 360 + 250 + 168 + 126 \ldots + 40 = 1339$, more than 2 times of Type A ($Q_{unit} = 440$). Discussing the advantages and disadvantages of Type A and Type B strategies as follow (in Table 3).

**Table 3.** The capacity and utilization of each rack in Type B ($Q_{rack} = 11$).

| Size No. | Capacity | Number of Cartons | $Utilization_{units}$ | $Utilization_{xy}$ |
|----------|----------|-------------------|-----------------------|---------------------|
| #1 | 360 | 49 | 13.61% | 13.61% |
| #2 | 250 | 44 | 17.60% | 17.60% |
| #3 | 168 | 53 | 31.55% | 28.86% |
| #4 | 126 | 43 | 34.13% | 30.58% |
| #5 | 96 | 43 | 44.79% | 38.70% |
| #6 | 90 | 37 | 41.11% | 41.11% |
| #7 | 65 | 50 | 76.92% | 67.22% |
| #8 | 48 | 47 | 97.92% | 75.20% |
| #9 | 44 | 44 | 100% | 82.62% |
| #10 | 40 | 40 | 100% | 87.11% |
| #11 | 40 | 40 | 100% | 100% |
| Total | 1339 | 490 | 36.59% | 52.96% |

1.  In Type A, 11 racks could content 440 cartons, no matter the sizes they are. In the worst situation, when the input cartons are all #11 size, Type A could be loaded 440 cartons, however Type B is 40. In brief, the utilization largely depends on the probability occurrence of the cartons' sizes.
2.  When the probability occurrence of 11 cartons' sizes is the same, the Type B's average capacity can over 460.

In Table 3, racks are loaded in 490 cartons, 4.5% more than Type A, then the $Utilization_{units} = 36.59\%$, and $Utilization_{xy} = 52.96\%$.

In fact, the rack-unit sizes in the manufacturing industry are limited, for too many different sizes of units are not convenient for sharing S/R machines, cranes, and lifters resources. Under this consideration, the loaded-units and cartons are simplified as three sizes, such as small, medium, and large. Using 80/20 law, the probability occurrence of small and medium cartons is nearly 80% in daily R/S activities, then we discuss the quantities and utilization of three racks strategies.

The procedure of Type B strategy is summarized as follow Algorithm 1.

---

**Algorithm 1.** The utilization of Type B strategy.

---

Input: $Q_{rack} = 11$, racks are empty. The quantities of small, medium and large racks are all at least 1.

      *qs*: The quantity of small racks, $\in [1,9]$. *qm*: The quantity of medium racks, $\in [1,9]$. *qb*: The quantity of large racks, $\in [1,9]$.

MAX: Making sure that no loaded-unit is available for a carton after loading cartons several times, setting MAX is more than $Q_{unit}$.

Procedure:

1: for *qs* = 1:9

2:    for *qm* = 1:9

3:          $qb = 9 - qm - qs$

4:          if $qb < 0$ break; end if

5:          calculate $Q_{unit}$ using Formula (4)

6:          for *i* = 1: MAX

7:             generate a random carton

                (the probability occurrences of small, medium and big are 20%, 60%, 20%)

8:             load carton into racks

9:             if (no loaded-unit is available for a random carton)

10:               calculate $Utilization_{units}$ and $Utilization_{xy}$, break;

11:            end if

12:        end for

13:    end for

14: end for

---

The process is shown in Algorithm 1, and using this algorithm, results are shown in Figure 10.

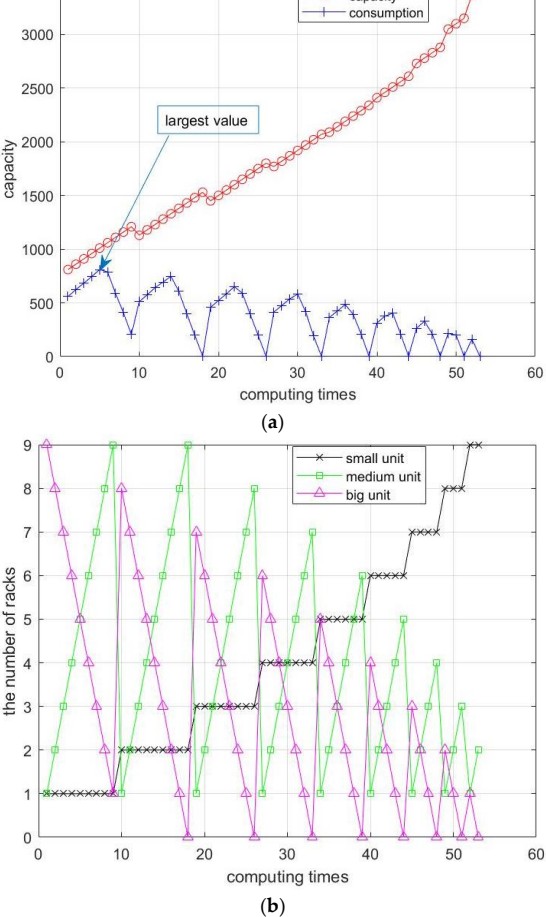

(a)

(b)

**Figure 10.** The capacity and consumption of Type B: (**a**) the capacity and consumption; and (**b**) the number of small, medium and big units.

In Figure 10, the consumption is largest when the small, medium, and big unit are 1, 5 and 5. At the same time, the capacity is 1010, and 812 units are loaded, then $Utilization_{units}$ = 80.4%, $Utilization_{xy}$ = 89.2%.

### 3.3. The Capacity and Utilization of Type C

In Type C, one rack-unit could be loaded several cartons, setting $Q_{rack} = 11$, and $Q_{unit} = 440$, using 80/20 law, we get $Q_{carton} = 525$, which represents that $Utilization_{units}$ is 119.3%, and $Utilization_{xy}$ is 60.6%. The results are better than Type A, but worse than Type B.

However, there are some disadvantages in the Type C strategy. For instance, multiple cartons in one unit, the single carton positions may be vague which brings difficulties in grabbing and loading requests. Otherwise, the target carton may be disturbed by nearby cartons and cause unsuccessfully grabs.

### 3.4. MRCGA Based on FF and BF Methods in Type A

When selecting loaded units to store cartons, we use FF and BF methods to compare their travel distance.

In Type A, setting $Q_{rack} = 2$, and units in rack#1 are numbered from 1 to 40, and 41 to 80 in rack #2. Two racks share a lane and a S/R machine to accomplish all storage and retrieval requests. Racks are half full of cartons under random loading, then input and output lists are randomly generated, as in the examples that follow.

The input list is (2 2 3 2 3 2 3 3 1 1 1 1 2 3 3 2 3 3 1 1), the numbers are matched with small, medium, and large cartons, for instance, "2" means a medium carton and could be loaded into a medium rack-unit, and "1" means a small carton.

The output list is (26 24 5 20 36 33 28 6 15 3 35 3 11 5 19 16), the numbers are the locations of cartons stored in racks, for instance, "26" means the carton stored in row 2, column 6 of rack #1.

In DC process, we obtain the optimal combination by MRCGA, which is shown in Figure 11 (Setting as follows, the generation is 99, population size is 300, *pc* is 0.9, and *pm* is 0.2).

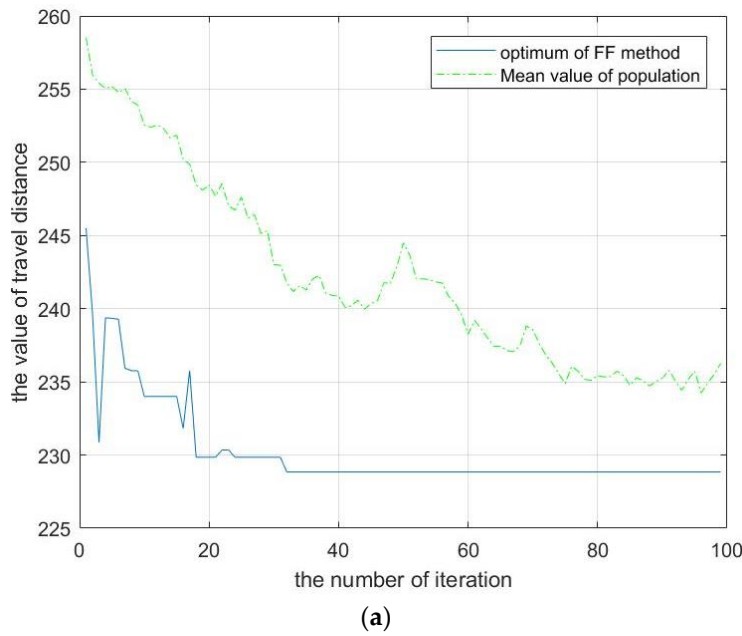

(**a**)

**Figure 11.** *Cont*.

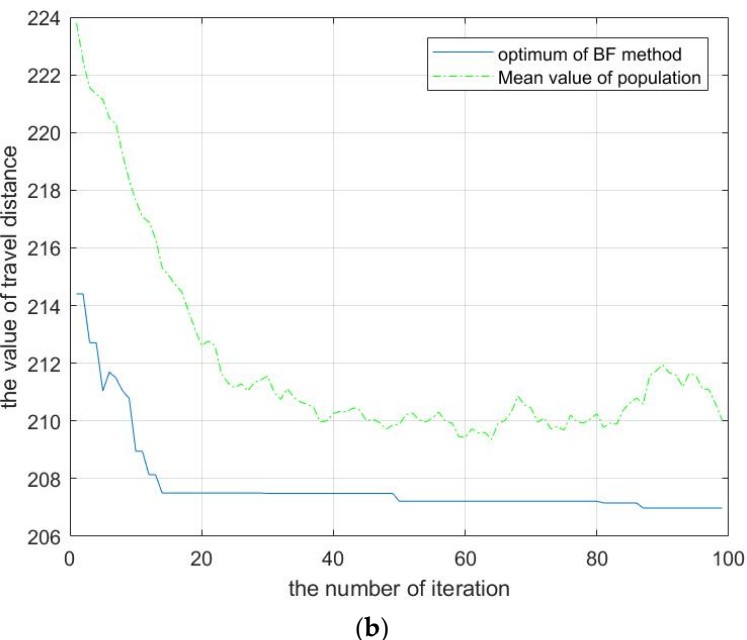

(**b**)

**Figure 11.** The FF and BF methods in Type A. (**a**) The FF method; (**b**) The BF method.

In Figure 11, the minimum travel distance is nearly 205 (the optimum of BF method line shows at the end of iteration) using BF method, obviously 228 using FF method, these represents a reduction of 11.2% in travel distance (Tokola and Niemi [26], optimization of the rule gaps are narrower than 10%). It is validated that MRCGA has an obvious advantage in solving DC problems. At the same time, one of the optimal solutions is shown in Formula (14).

$$Chr_{optimal,BF} = \begin{bmatrix} 2 & 0 \\ 2 & 3 \\ 3 & 28 \\ 2 & 43 \\ 3 & 20 \\ 2 & 51 \\ 3 & 0 \\ 3 & 59 \\ 1 & 0 \\ 1 & 26 \\ 1 & 24 \\ 1 & 5 \\ 2 & 0 \\ 3 & 33 \\ 3 & 15 \\ 2 & 56 \\ 3 & 75 \\ 3 & 36 \\ 1 & 45 \\ 1 & 6 \end{bmatrix} \tag{14}$$

As shown in Formula (14), $Chr_{optimal,BF}$ is a optimal solution by BF method. On the right-hand side, the first column is the input requests, and the second is output requests, and they are combined in rows, the DC cycles. Actually, the travel distance is 205.51 of this solution.

### 3.5. MRCGA Based on Three Types Rack Strategies

In practical application, the capacity of an Miniload AS/RSs could reach 100,000, and the daily R/S requests are up to 2000 [16]. Based on three types of racks strategies, Setting $Q_{rack} = 22$ (In Type B, $Q_{rack,small} = 2$, $Q_{rack,medium} = 10$, $Q_{rack,large} = 10$). At the same time, the racks are half full of random cartons, assuming that an input list contains 100 storage cartons, and an output list contains 80 retrieval requests, considering the inputs are more than outputs, there is a situation that no unit is available for cartons after multiple cycles, then we compare the utilization of three rack strategies by FF and BF methods using MRCGA.

The procedure of Type A-FF strategy is summarized as follow Algorithm 2, and an example of $Map_{units}$ is showed in Figure 12.

---

**Algorithm 2.** The utilization of Type A-FF strategy using MRCGA.

---

Input: $Q_{rack} = 22$, setting that the quantities of inputs and outputs are 100 and 80.

Racks are half full, and $Map_{units}$ is the racks map, which shows there are or not cartons in units. When one unit is loaded with some carton, the unit value in $Map_{units}$ is setting as carton length; when one carton is picked and removed, the unit value in $Map_{units}$ is setting as "0" (in Type A and Type B) or "1"/"2"/"3" (in Type C). In Type A model, one unit is loaded only one carton, the unit value in $Map_{units}$ is setting as "1" or "0". In Type B model, one unit is loaded only one carton in racks, no matter what rack-unit size it is, the unit value in $Map_{units}$ is setting as "1" or "0". In Type C model, one unit could be stored several cartons, and the unit value in $Map_{units}$ is added or subtracted, and when one unit is loaded with a carton, the unit value add the carton's length, if the added unit value is more than unit length, this unit cannot be loaded with this carton, and the program runs to next unit.

As shown in Figure 12, the first column is number of units:

The value is $(1, 2, \ldots ,Q_{rack,1}, Q_{rack,1} + 1, Q_{rack,1} + 2, \ldots , Q_{rack,1} + Q_{rack,2}, \ldots , Q_{rack,1} + Q_{rack,1} + \ldots + Q_{rack,N} \ldots )$, N is the total number of racks.

The second column is unit length, the value is 3 in Type A and Type C, and 1, 2, or 3 in Type B.

The last column is unit value, "0" is empty, "1"/"2"/"3" are the length of loaded cartons, and the gaps in units are equal to unit length subtract unit value.

EI: estimate of iterations, such as "30", making sure a situation that no unit is available for cartons after several iterations. Then, the capacity of racks reaches a maximum, calculating $Utilization_{units}$ and $Utilization_{xy}$. Usually, calculating multiple times (MaxCalculateTime is more than 20) for average values.

Procedure:

1: for $i = 1 : MaxCalculateTime$

2:    function Type A_FF:

3:        for $j = 1 : EI$

4:            generate randomly inputs and outputs lists,
                InputList (quantity is 100), OutputList (quantity is 80), by 80/20 law.

5:            generate initial population, chromosomes are combined randomly.
                use MRCGA, get $Chr_{optimal,FF}$, update the $Map_{units}$.

6:            if (no unit is available for a carton)

7:                get the quantity of cartons in $Map_{units}$, and calculate $Utilization_{units}$ and $Utilization_{xy}$

8:                break

9:            end if

10:        end for

11: end for

12: calculate the average of $Utilization_{units}$ and $Utilization_{xy}$.

---

Taking average values after multiple computing, we get the utilization of 6 strategies, as shown in Table 4.

| Number of units | unit length | unit value |
|---|---|---|
| 1 | 3 | 0 |
| 2 | 3 | 0 |
| 3 | 3 | 0 |
| 4 | 3 | 0 |
| 5 | 3 | 0 |
| 6 | 3 | 0 |
| 7 | 3 | 0 |
| 8 | 3 | 0 |
| 9 | 3 | 0 |
| 10 | 3 | 0 |
| 11 | 3 | 0 |
| 12 | 3 | 0 |
| 13 | 3 | 0 |
| 14 | 3 | 0 |
| 15 | 3 | 0 |
| 16 | 3 | 0 |

**Figure 12.** An example of $Map_{units}$.

**Table 4.** The capacity and utilization of DC using MRCGA.

| Name | Capacity | $Utilization_{units}$ | $Utilization_{xy}$ |
|---|---|---|---|
| Type A-FF | 880 | 100% | 46.22% |
| Type A-BF | 880 | 100% | 46.15% |
| Type B-FF | 2020 | 93.76% | 98.21% |
| Type B-BF | 2020 | 93.09% | 98.19% |
| Type C-FF | 880 | 170.85% | 80.41% |
| Type C-BF | 880 | 201.05% | 99.68% |

In Table 4, Type A has the least capacity, compared to others, and Type C has the most. FF and BF methods are given nearly same utilization values in Type A and Type C, thus BF is nearly 20% better than FF in Type C. The units in Type B-FF, Type B-BF and Type C-BF methods are nearly full loaded, showing an excellent utilization.

## 4. Conclusions

In a limited space, choosing an appropriate rack strategy largely depends on the containers' sizes and probabilities. In this paper, we discussed three rack strategies in general applications, such as that the cartons had equal probabilities or by the 80/20 law, comparing their capacities, utilization of units and space. In the storage and retrieval process, we provided FF and BF methods to locate cartons, and used an MRCGA to generate the optimal solution of DC cycles and list orders. The results showed that BF represents a reduction of 11.2% in travel distance, obviously improving productivity. The capacity of Type B is more than two times that of Type A or Type C, and its utilization is relatively excellent, just 1.47% less than Type C. In Type C, the BF method showed a greater advantage than FF, adding nearly 20% utilization in space. From this analysis, we can draw some management implications: storage requirements are changing but can be foreseeable; considering their advantages and disadvantages, Type B-FF, Type B-BF and Type C-BF could better meet high-density requirements, and provide diversified storage and retrieval solutions for Manufacturing Enterprises in Miniload AS/RSs.

However, the Type A rack design is still typical in our life, and Type B and Type C have their limits in practical applications; for example, Type B is useful when input carton sizes are foreseeable, or the unforeseen changes of carton sizes lead to unsuitable storage units and a greater waste of space. Type C has high requirements for the accurate calculation of carton locations. In addition, three models were established relating to two directions,

length and height, and the third direction (depth) was supposed as equal, greatly reducing the amount of calculation but showing some discrepancies from the reality. In the future, we will consider improving the models in these directions.

**Author Contributions:** Conceptualization, M.H. and Z.G.; methodology, M.H.; software, M.H.; validation, M.H., C.W. and G.H.; formal analysis, M.H.; investigation, M.H.; resources, M.H.; data curation, M.H.; writing-original draft preparation, M.H.; writing-review and editing, C.W.; visualization, M.H.; supervision, Z.G. and G.H.; project administration, M.H.; funding acquisition, Z.G. and G.H. All authors have read and agreed to the published version of the manuscript.

**Funding:** This work was funded by the National Natural Science Foundation of China, grant number 51979115 and grant number 51679099.

**Data Availability Statement:** Not applicable.

**Conflicts of Interest:** The authors declare no conflict of interest.

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
