# Peer review of "Multiple-Rack Strategies Using Optimization of Location Assignment Based on MRCGA in Miniload Automated Storage and Retrieval System"

_processes, doi:10.3390/pr11030950_

Round 1

Reviewer 1 Report

The research work presented by the authors is well structured and ordered, with well-done, clear, allusive, pertinent and well-explained figures. 

Without detriment to the foregoing, authors are recommended to take into account the following typographical criteria:

1) Lines 5 through 10 do not indicate who is or are the correspondence author(s).

2) Throughout the entire text, the authors do not include a space between the end of a word and the beginning of a parenthesis. For example, on line 13, “…Systems(AS/RS)…”, should be written as “…Systems (AS/RS)…”. On line 25, “…rise[1],…”, should be written as “…rise [1],…”. On line 48, “…Roodbergen et al.[7]…” should read “…Roodbergen et al. [7]…”.

3) In all cases, when a word or phrase ends in a point (.) or a comma (,), there must be a space before the next word, number, or symbol. The omission of this space is a recurring typographical error in the manuscript. An example of the above can be found in line 37, “…Fig.1…”, should be written as “…Fig. 1…".

4) I recommend that the expressions: TypeA, TypeB, and TypeC, be written as Type A, Type B, and Type C, respectively.

5) Authors should review the consecutive numbering of the pages. There are some of them that are not numbered. There are also pages with the same numbering.

6) I suggest that, in the equations, instead of using the multiplication symbol “*”, change it to the symbol “∙”.

7) When citing an equation in the text of the manuscript, it is recommended to use “Equation (X)” or “Equation X”, instead of “Formula.X”.

8) In Figures 2 and 3, authors enter the terms “Rack 1#, Rack 2#, Rack N#”. These are later used in the document. I think the correct way is “Rack #1, Rack #2, Rack #N”.

9) Check the numbering of the figures, there are two of them with the number 3, then jump from 5 to 7. Check the citation in the text of those figures that will change their numbering.

10) Under the figures, in several cases after the point followed by the number of the figure, the title of the figure does not begin with a capital letter. For example, the manuscript says “Figure 2. the side view of TypeA racks.”, but it should say “Figure 2. The side view of Type A racks.”. This must be checked and corrected in all figures.

11) On line 226 of the manuscript it says “…law([39] Zhang, X. , et al), the…”. I suggest changing to “…law (Zhang, X., et al. [39]), the…” or simply “……law [39], the…”.

12) On line 235 there is a text with additional information about one of the graphs in Figure 10. I recommend that this text not overlap the figure. In the same figure, I recommend denoting each graph as (a) and (b), indicating this and its content on line 237, for example, something like Figure 10. Numerical results from strategy Type B. (a) Capacity, (b) Consumption .

13) On line 292 it says “…TypeA and TypeC…”, I think it should say “…Type A and Type B…”.

Reviewer 2 Report

please check the attached file

Reviewer 3 Report

The article touches on optimizing material handling systems in distribution warehouses, which, with a shortage of employees, especially in recent years, use high-performance automatic solutions. The article has a scientific value and contains an element of novelty, although the presented solutions will not be groundbreaking. The Authors should consider the following remarks:

Miniload and installation based on shuttle devices are usually considered different systems. Miniload basically uses stacker cranes for containers. Consider changing the study title and selected passages to eliminate potential confusion.

Roodbergen or van den Berg are classic elements of a literature review for this topic. However, the problem of rationalization of transport cycles in shuttle systems has been considered in detail in recent years. Authors should focus their literature review on papers that detail the issues they deal with. The overview in its current form is quite broad but superficial.

Table 1 should be in the literature review section.

Authors correctly formulated the assumptions for the model.

The mathematical model was formulated correctly, although incorrect symbols were used in places (e.g. "*" instead of the multiplication sign), and the way of defining the indexes is unclear. The authors should emphasize more the functions of the criterion of individual scenarios.

Figure 7 presents quite obvious assumptions of the method.

The MRCGA algorithm was used correctly.

Round 2

Reviewer 2 Report

The paper can be accepted